



# Effect of OH radiance on the temperature and wind measurements derived from limb viewing observations of the 1.27 μm O₂ dayglow

Kuijun Wu[1], Weiwei He[2], Yutao Feng[3], Yuanhui Xiong[1], Faquan Li[1]

[1]State Key Laboratory of Magnetic Resonance and Atomic and Molecular Physics, Wuhan Institute of Physics and Mathematics, Chinese Academy of Sciences, Wuhan National Laboratory for Optoelectronics, Wuhan 430071, China.
[2]City College, Wuhan University of Science and Technology, Wuhan, Hubei, 430083, China.
[3]Key Laboratory of Spectral Imaging Technology of Chinese Academy of Sciences, Xi'an Institute of Optics and Precision Mechanics, Chinese Academy of Sciences, Xi'an 710119, China.

*Correspondence to*: Kuijun Wu (wukuijun@wipm.ac.cn)

**Abstract.** The $O_2(a^1\Delta_g)$ emission near 1.27 μm is well-suited for remote sensing of global wind and temperature in near-space by limb-viewing observations due to its bright signal and extended altitude coverage. However, vibrational-rotational emission lines of the OH dayglow produced by the Hydrogen-Ozone reaction ($H + O_3 \rightarrow OH^* + O_2$) overlap the infrared atmospheric band emission ($a^1\Delta_g \rightarrow X^3\Sigma_g$) of $O_2$. The main goal of this paper is to discuss the effect of OH radiance on the wind and temperature measurements derived from the 1.27 μm $O_2$ dayglow limb-viewing observations. The $O_2$ dayglow and OH radiance spectrum over the spectral region and altitude range of interest is calculated by using the line-by-line radiative transfer model and the most recent photochemical model. The method of four-point sampling of the interferogram and sample results of measurement simulations are provided for both $O_2$ dayglow and OH radiance. It is apparent from the simulations that the presence of OH radiance as an interfering species decreases the wind and temperature accuracy at all altitudes, but this effect can be reduced obviously by improving OH radiance knowledge.

## 1 Introduction

The infrared atmospheric band emission ( $a^1\Delta_g \rightarrow X^3\Sigma_g$ ) of $O_2$ observed near the wavelength of 1.27 μm has remarkable advantages in atmospheric remote sensing due to its bright signal and extended altitude coverage (Mlynczak et al., 2007). The enormous success of Wind Imaging Interferometer (WINDII) (Shepherd et al., 2012), High Resolution Doppler Imager (HRDI) (Ortland et al., 1996) and TIMED Doppler Interferometer (TIDI) (Killeen et al., 2006), which measured winds in the upper mesosphere and lower thermosphere using Doppler shifts in visible airglow emission lines, stimulates interest in measuring wind and temperature from limb-viewing satellites using the 1.27 μm dayglow (Wu et al., 2018).

W. E. Ward *et al* designed a high sensitivity interferometer, WAMI, to provide simultaneous measurements of horizontal wind and rotational temperature by using the combination of a "strong" emission lines group and a "weak" group of the $O_2(a^1\Delta_g)$ airglow (Ward et al., 2001). A similar instrument, MIMI, designed by York University, also takes advantage of strong and weak emission lines for dynamics and thermodynamics measurement. Both WAMI and MIMI are imaging, field-



widened Michelson interferometers and the same measurement technique known as Doppler Michelson imaging interferometry is employed successfully by the WINDII instrument on NASA's UARS satellite (Shepherd et al., 2012). The observing strategy of using two sets of three emission lines makes it a perfect approach for WAMI and MIMI to improve our knowledge of the wind and temperature of the lower thermosphere and middle atmosphere, as well as global distribution and

transport of $O_3$. In recent years, measuring wind and temperature in the planetary atmosphere such as Mars and Venus is also proposed by using 1.27 µm band emission of the electronically excited $O_2(a^1\Delta_g)$ state (Ward et al., 2002; Zhang et al., 2017). Closely following the WAMI concept, we recently proposed a near-space wind and temperature sensing interferometer (NWTSI) for simultaneous measurements of the atmospheric wind and temperature in the near space from the limb-viewing satellite by observing the $O_2(a^1D_g)$ dayglow near 1.27 µm (He et al., 2019).

The strategy of choosing two sets of three emission lines adopted by WAMI, MIMI and NWTSI is quite ingenious. Three emission lines of each group are relatively well separated and belong to different branches within the same band, which allows them to be optically isolated and ensures distinct temperature sensitivities. Additionally, the line strengths of strong group and the weak one differ by about one order of magnitude, so their radiation intensities and absorption characteristics varying with altitude are available for covering a great extended altitude range from 45 to 100 km.

However, vibrational-rotational emission lines of the OH dayglow produced by the Hydrogen-Ozone reaction (H + $O_3 \rightarrow$ OH* + $O_2$) overlap the infrared atmospheric band emission ($a^1\Delta_g \rightarrow X^3\Sigma_g$) of $O_2$ near 1.27 µm (Maihara et al., 1993), which will surely contribute to the increase of wind and temperature errors.

The main goal of this paper is to discuss the effect of OH radiance on the wind and temperature measurements derived from the 1.27 µm $O_2$ dayglow limb-viewing observations. The atmospheric radiance model is simulated by line-by-line radiative

transfer method. A brief description of $O_2$ dayglow and OH radiance spectrum over the spectral region and altitude range of interest is provided. The forward simulation including instrument model and the method of four-point sampling of the interferogram, as well as sample results of measurement simulations is presented. Inversion errors of wind and temperature measurements due to the effect of OH radiance are presented and discussed. Here we report, to the best of our knowledge, the first consideration and discussion of the effect of OH radiance on the temperature and wind measurements using the 1.27 µm

$O_2$ dayglow.

## 2 O₂ dayglow and OH radiance spectrum

### 2.1 The O₂(a¹Δg) and OH dayglow volume emission rates

Two predominant mechanisms, the photolysis of $O_3$ and the energy transfer from $O(^1D)$, produce the lowest-lying electronic state of molecular oxygen $O_2(a^1\Delta_g)$ in the mesosphere and lower thermosphere. $O_2(a^1\Delta_g)$ radiates strong emission in the





infrared atmospheric band $O_2(a^1\Delta_g) \to O_2(X^3\Sigma_g)$ and produces dayglow at 1.27 µm. The volume emission rate (VER) of a

O₂ individual rotational line, $\eta_{O_2,rot}$, can be given by (Mlynczak et al., 1993)

$$\eta_{O_2,rot} = A \frac{g'}{Q_{O_2}(T)} \exp\left(\frac{-hcE'}{kT}\right) \frac{\phi_\alpha R_1[O_3] + \sum_{i=1}^{5} K_i[Y_i] \dfrac{[O_2]\left\{R_2 + \phi_\eta R_3[O(^1D)]\right\}}{A_{O_2(b^1\Sigma_g)} + \sum_{i=1}^{5} K_i[Y_i]}}{A_{O_2(a^1\Delta_g)} + \sum_{i=1}^{3} C_i[X_i]} \tag{1}$$

where $A$ is the Einstein coefficient of the transition, $g'$ is the upper state degeneracy, $Q(T)$ is the rotational partition function,
$h$ is the Planck constant, $c$ is the light speed, and $k$ is the Boltzmann constant, $T$ is the rotational temperature, $E'$ is the upper
state energy, $X = \{O_2, N_2, O\}$, $Y = \{N_2, O_2, CO_2, O_3, O\}$, $R_1 = 8.1 \times 10^{-3}$, $R_2 = 5.35 \times 10^{-9}$, $R_3 = 3.2 \times 10^{-11} \exp(70/T)$
$C_1 = 3.6 \times 10^{-18} e^{-220/T}$, $C_2 = 1.0 \times 10^{-20}$, $C_3 = 1.3 \times 10^{-16}$, $K_1 = 2.1 \times 10^{-15}$, $K_2 = 4.2 \times 10^{-13}$, $K_3 = 2.2 \times 10^{-11}$, $K_4 = 8.0 \times 10^{-14}$ and
$K_5 = 3.9 \times 10^{-17}$, $0.54 < \phi_\eta < 1.0$.

    An enhanced concentration of OH* in high states occurs in a thin layer near the mesopause due to the Hydrogen-Ozone
reaction, $H + O_3 \to OH^* + O_2$. A multitude of excited OH* in high vibrational and rotational states cascading to lower energy
states results in an emission spectrum, which extends over a wide wavelength range (500–5000 nm). Among those vibrational-
rotational transitions, two emission lines, RR$_{2.5e}$ and RR$_{2.5f}$ of the $OH(X^2\Pi_{3/2}, \upsilon' = 8) \to OH(X^2\Pi_{3/2}, \upsilon'' = 5)$ band, are very
close to the three weak target emission lines of the 1.27 µm O₂ dayglow.

    The VER of the OH (8–5) vibrational band can be given by (Russell and Lowe, 2003)

$$\eta_{OH,8-5}(J', J'') = 2(2J'+1) \frac{[OH(v=8)]A_{8-5}^{J',J''}}{Q_{OH}(T)} \exp\left(-\frac{hcF_{8-5}(J')}{kT}\right) \tag{2}$$

where $[OH(v=8)]$ is the number density of OH in the $v = 8$ state, $A_{8-5}^{J',J''}$ is the particular Einstein coefficient for the rotation
transition of the (8–5) vibrational band, $J'$ and $J''$ are initial and final values of the total angular momentum of the transition,
$F_{8-5}(J')$ is the upper state rotational term value.

    Figure 1 shows the VER profiles of both the three weak target emission lines of the 1.27 µm O₂ dayglow and the two
emission lines, RR$_{2.5e}$ and RR$_{2.5f}$ of the $OH(X^2\Pi_{3/2}, \upsilon' = 8) \to OH(X^2\Pi_{3/2}, \upsilon'' = 5)$ band. As can be seen, the O₂ dayglow
peaks at about 45 km and the OH reaches 90 km, and the emission rate of O₂ is roughly 45 times stronger than OH. In addition,
the two emission lines of OH show a similar distribution profile over altitude, while, the three emission lines of O₂ differs a
lot. This is because the two emission lines of OH share the same particular Einstein coefficient and total angular momentum,
but the three emission lines of O₂ are located in different vibration bands with sufficiently different lower-state energy, which
leads to a disparity in temperature sensitivity.




## 2.2 The O₂(a¹Δₘ) and OH dayglow limb radiance

In the case of limb-viewing, each viewing direction defines a ray path. The path segments defined by the intersection of atmospheric layers and ray paths is assumed to have the same emission and absorption characteristics. Evaluated on the layer-by-layer basis, the observed spectral irradiance is considered as a path integral along the line of sight (Song et al., 2017)

$$L(\nu) = \int_{-\infty}^{\infty} \eta(s) D(\nu, s) \exp[-\int_{-s}^{\infty} n(s')\sigma(s') \, ds'] \, ds \tag{3}$$

where $D(\nu)$ is the Doppler line shape of the spectral line, $\eta(s)$ is the VER, $n(s)$ is the number density, $\sigma(s)$ is the absorption cross-section and $s$ is the distance along the line-of-sight.

Figure 2 shows the limb spectral radiance of three emission lines of $O_2$ and the two emission lines of OH at different altitudes (40-90 km with 10 km interval).

The total radiance including limb spectral radiance of the weak group of $O_2$ dayglow and the OH radiance is shown in Fig. 3. As can be seen, the third emission line of the $O_2$ dayglow near 7823 cm⁻¹ is too closed to the OH lines RR$_{2.5e}$ and RR$_{2.5f}$ (less than 0.05 nm) to be well optically isolated. The OH radiance will surely affect the spectral integral intensity of the $O_2$ mission line near 7823 cm⁻¹ especially for altitudes between 80 to 90 km where the OH radiance is relatively strong.

## 3 Forward simulation

### 3.1 The instrument model

The NWTSI using the combination of a "strong" emission lines group and a "weak" group of the $O_2(a^1\Delta_g)$ airglow is a limb-viewing satellite instrument (He et al., 2019). The schematic drawing of Limb-imaging geometry and Instrument concept is shown in Fig. 4, which closely follows the WAMI concept (Ward e et al., 2001). The field of view (FOV) of NWTSI is defined by the first telescope with a value of 1.5° × 1.5°, which allows NWTSI covering an entire altitude range from 20 to 120 km in single images for a nominal spacecraft altitude of 650 km.

The entrance aperture of NWTSI is about 5 cm in diameter with an effective aperture ratio of f/1.3. The magnifications of the first telescope and second are 2 and 0.5 respectively, so the FOV at the Michelson is 3° × 3° and at the filters is again 1.5° ×1.5°. The beam splitter of the NWTSI field-widened Michelson interferometer consists of two cemented half hexagons with a low-polarizing semireflecting dielectric multilayer on one of the diagonal faces. It is made of BK7 glass and the entrance and exit faces are 6.35 cm². Large optical path difference and field widening involve using 12.240 cm LaKN12 glass and 11.070 cm BK7 glass with corresponding refractive indices 1.733 and 1.516 in the two arms of the interferometer. In order to avoid errors caused by intensity variations during measurements, the four interferogram samples are taken simultaneously rather than sequentially by dividing one Michelson mirror into four equal segments so the steps are coated permanently onto the mirror. A composite of etalon and interference filter is used to isolate individual emission lines. The etalon is fused silica





with a finesse of 20 and a free spectral range of 2.0 nm. The etalon thickness is designed to give optimum transmittance in the outer one-third of the field of view for three strong/weak O2 emission lines. The imaging system produces four copies of FOV by using a shallow, pyramid shaped prism just in front of the camera. The edges of the prism are aligned with the divisions between the quadrants of the sectored Michelson mirror to form 4 images on the array detectors simultaneously, corresponding different sampling steps of the Michelson segments.

**3.2 The imaging interferogram**

By sampling the interferogram for each pixel at four points corresponding to the four phase steps, the Doppler wind is obtained, which is so called the four-point sampling method. The equation representing the interferogram for a given pixel at row $l$ and column $j$ of the detector is (Rahnama et al., 2006)

$$I_{klj} = R_{lj} \int_{v_1}^{v_2} f_{lj}(v) \cdot L_{lj}(v) \cdot \left[1 + U_{lj} \cos(2\pi v \Delta_{lj} + \varphi_{klj})\right] dv \qquad (4)$$

where $f_{lj}(v)$ is the relative total filter function, $U_{lj}$ is the instrument visibility, $\Delta_{lj}$ is the OPD, and $\varphi_{klj}$ is the Michelson interferometer $k$th phase step, $v$ is the wavenumber, the instrument responsivity $R_{lj}$ is defined by

$$R_{lj} = \frac{A\Omega t \tau q}{h c v_0} \qquad (5)$$

where $A\Omega$ is the Pixel etendue, $t$ is the Exposure time, $\tau$ is the attenuation, $q$ is the quantum efficiency.

The simulated mean value of the interferogram for the weak group of $O_2$ dayglow as well as the OH radiance is shown in

Fig. 5. The pattern on Fig. 5(a) reflects the variation over the field of the filter transmittance function, the dependence of optical path difference on pixel positions, and the tangent height variation of $O_2$ dayglow within a column. The image in Fig. 5(b) shows the interference fringe of OH radiance is much stronger for pixels nearer the periphery of FPA, where the signal of the third weak emission line of the $O_2$ dayglow near 7823 cm$^{-1}$ is imaged.

**4 Inversion error due to the effect of OH radiance**

Molecular species in NWTSI's selected spectral range of 7820-7824 cm$^{-1}$ and 7908-7912 cm$^{-1}$ other than $O_2$ can affect NWTSI's wind and temperature measurements through absorption and emission. Limb spectral radiance of OH airglow is the most important interfering constituents for NWTSI, especially for the "weak" group detection, as shown in Fig. 3 and Fig. 5. Using etalon as the ultra-narrow filter significantly reduces but does not eliminate the influence from OH airglow. Therefore, the uncertainty in its mixing ratio surely contributes to the increase of wind and temperature errors.

The wind velocity $v_w$ is measured as a phase shift $\delta\varphi$ of the interferogram





$$v_w = \frac{c}{2\pi\Delta v_0}\delta\varphi \tag{6}$$

And phase $\varphi$ can be calculated from:

$$\varphi_t = \tan^{-1}(\frac{J_3}{J_2}) \tag{7}$$

Fourier coefficients, $J_1$, $J_2$, and $J_3$, also referred to as the apparent quantities, are related to any point $k$ along a fringe
interferogram $I$ (Shepherd et al., 2012):

$$J_1 = I_{mean} = \frac{1}{4}\sum_{k=1}^{4} I_k$$
$$J_2 = \frac{1}{2U}\sum_{k=1}^{4} I_k \cos(\varphi_{klj}) \tag{8}$$
$$J_3 = \frac{1}{2U}\sum_{k=1}^{4} I_k \sin(\varphi_{klj})$$

Omitting the presence of OH emission will change the value of $J_2$ and $J_3$, which will inevitably lead to an inversion error
for the wind measurement. The contribution of OH radiance to the wind error is shown in Fig. 6(a). As can be seen, the wind
error is about 1-2 m/s for altitudes below 70 km, climbing to greater than 8 m/s as the altitude increases from 80 to 90 km. The
sensitivity of wind retrieval is weaker in the lower altitudes due to lower OH volume emission rate. A similar trend is found
for the temperature obtained from $J_1$: the temperature error is about 4-6 K for altitudes below 70 km and greater than 20 K for
altitudes from 80 to 90 km (see Fig. 6(b)). By happy coincidence, the wind and temperature inversion for altitude above 70
km is irrelevant to the "weak" group. The three strong emission lines suffering little from self-absorption at relatively higher
tangent heights takes the place of the weak group for data inversion.
The effect of OH airglow on wind and temperature inversion can be reduced by excluding the tainted data pixels or
improving OH radiance knowledge. The first option is infeasible for NWTSI because the weak emission line tainted by the
OH airglow has the largest spatial coverage. Another option is to retrieve OH radiance. This option might potentially improve
OH radiance knowledge to a very limited degree (depending on the uncertainty of the initial guess), and resulting OH radiance
error will also be introduced by the retrieval to compensate for other error sources that are not well accounted for.
The degree of knowledge of OH radiance is important for NWTSI retrieval. Fig. 7 shows the errors deviation of inverted
wind and temperature with OH radiance knowledge of 80 % and 90 %. As can be seen, when OH radiance is known to 80%,
the wind error is 0.3–0.5 m/s and temperature error is 0.7-1.0 K for the altitude range of 40–70 km; for OH radiance knowledge
of 90%, the wind error decreases to 0.1–0.3 m/s m and temperature error decreases to 0.3-0.5 K.
The performance analyses presented assume that the OH radiance is known to a degree that results in meeting the threshold
wind and temperature accuracies. The largest errors are obtained when not accounting for OH radiance in the retrievals. In



order to ring down the wind and temperature error to an acceptable level, OH radiance knowledge with an accuracy of 80-90 % is required.

## 5 Conclusions

We have simulated and discussed the effect of OH radiance on the wind and temperature measurements derived from the 1.27 μm $O_2$ dayglow limb-viewing observations. We first calculated the $O_2$ dayglow and OH radiance spectrum over the spectral region and altitude range of interest using the line-by-line radiative transfer model and the photochemical model incorporating the most recent spectroscopic parameters, rate constants and solar fluxes. Its shows that the OH lines $RR_{2.5e}$ and $RR_{2.5f}$ is too closed to the third weak emission line of the $O_2$ dayglow near 7823 cm$^{-1}$ (less than 0.05 nm), which will surely affect the spectral integral intensity of the $O_2$ mission line. A brief description of the instrument model of NWTSI including
the schematic drawing of Limb-imaging geometry and the instrument concept is presented. The method of four-point sampling of the interferogram and sample results of measurement simulations are provided for both $O_2$ dayglow and OH radiance. It was apparent from the simulations that the interference fringe of OH radiance is much stronger for pixels nearer the periphery of FPA, where the signal of the third weak emission line of the $O_2$ dayglow near 7823 cm$^{-1}$ is imaged. Inversion errors of wind and temperature measurements due to the effect of OH radiance are presented and discussed in detail. The presence of OH
radiance as an interfering species decreases the NWTSI performance at all altitudes with the largest impact especially for altitudes from 80 to 90 km. The effect of OH airglow on NWTSI inversion can be reduced by improving OH radiance knowledge. Accurate $OH(v = 8)$ concentrations (uncertainty level of 20 % or better) are required to help meet the wind and temperature accuracy requirements.

## Acknowledgments

This work was supported by National Natural Science Foundation of China (NSFC) (61705253) and National Key R&D Program of China (2017YFC0211900). The authors gratefully acknowledge informative and helpful discussions with Wang Dingyi on theoretical details of this work.

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

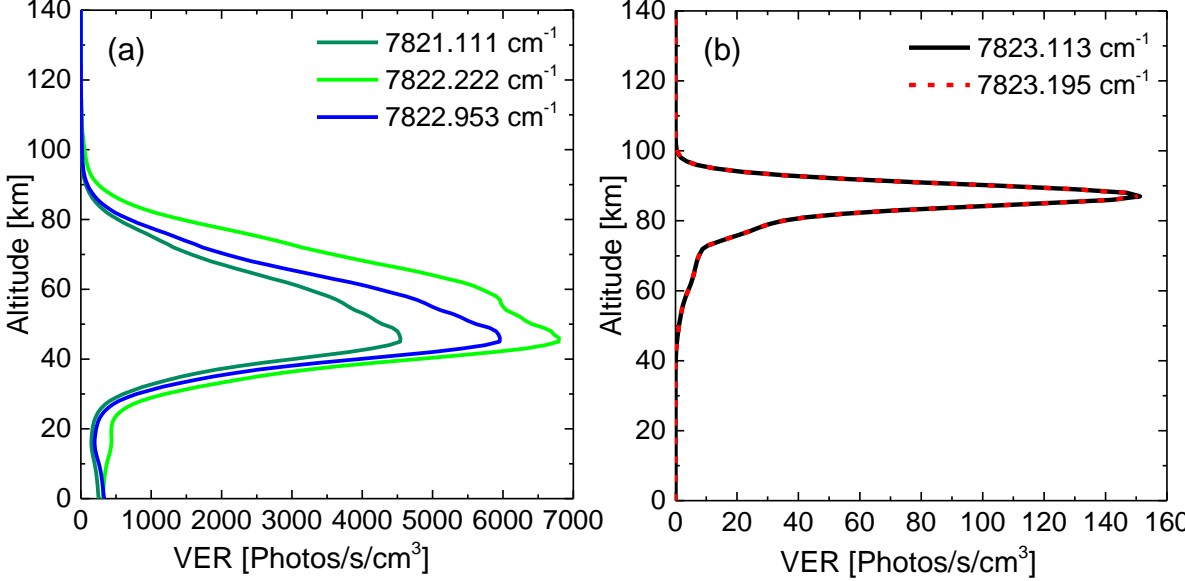

**Figure. 1: The VER profiles of both the three weak target emission lines of the 1.27 μm O$_2$ dayglow and the two emission lines of OH.**





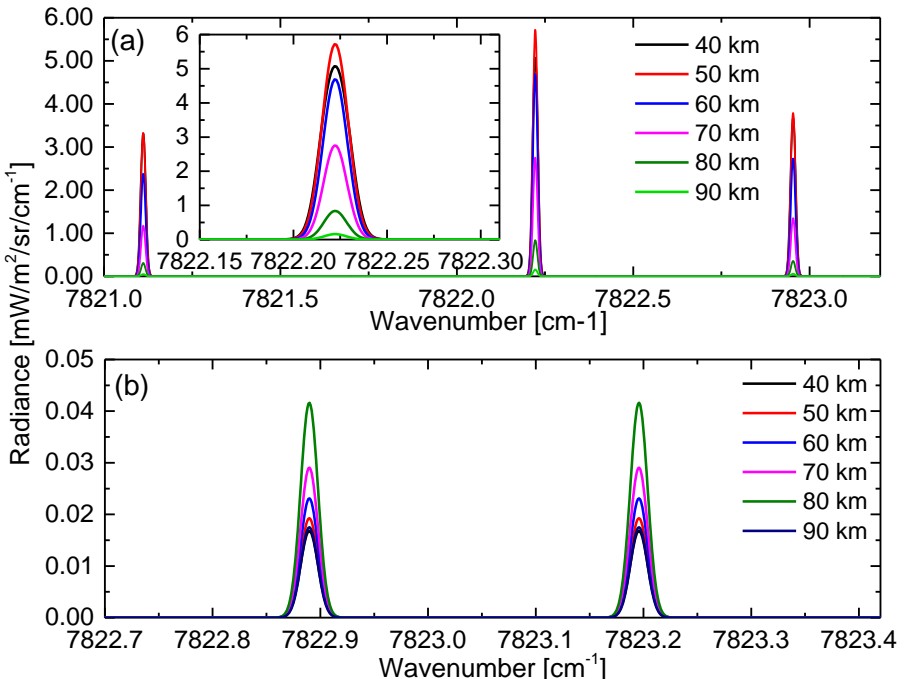

**Figure. 2: The emission spectra of the three emission lines of O2 dayglow and the two emission lines of OH at tangent heights of 40 km, 50 km, 60 km, 70 km, 80 km and 90 km.**

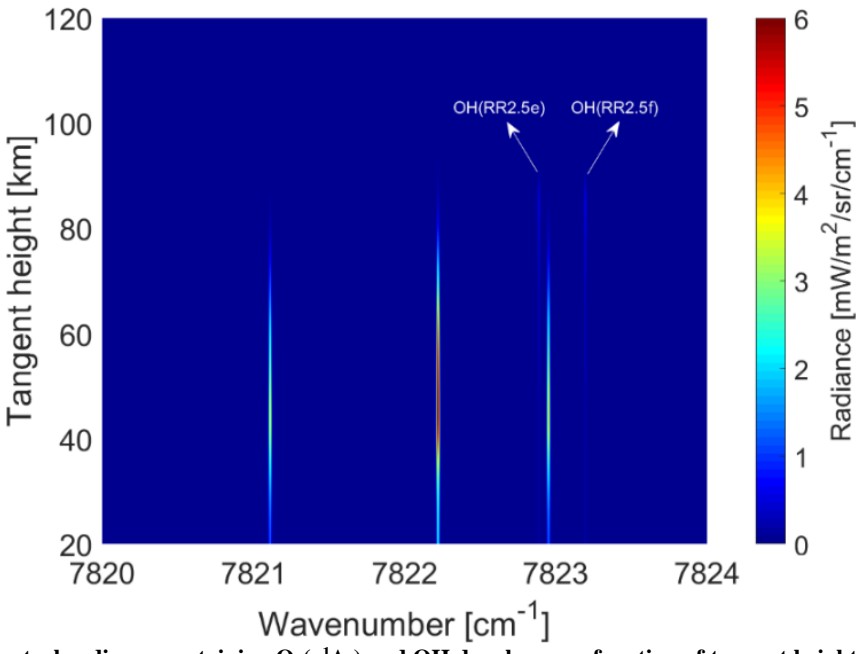


**Figure. 3: The total spectral radiance containing $O_2(a^1\Delta_g)$ and OH dayglow as a function of tangent height.**





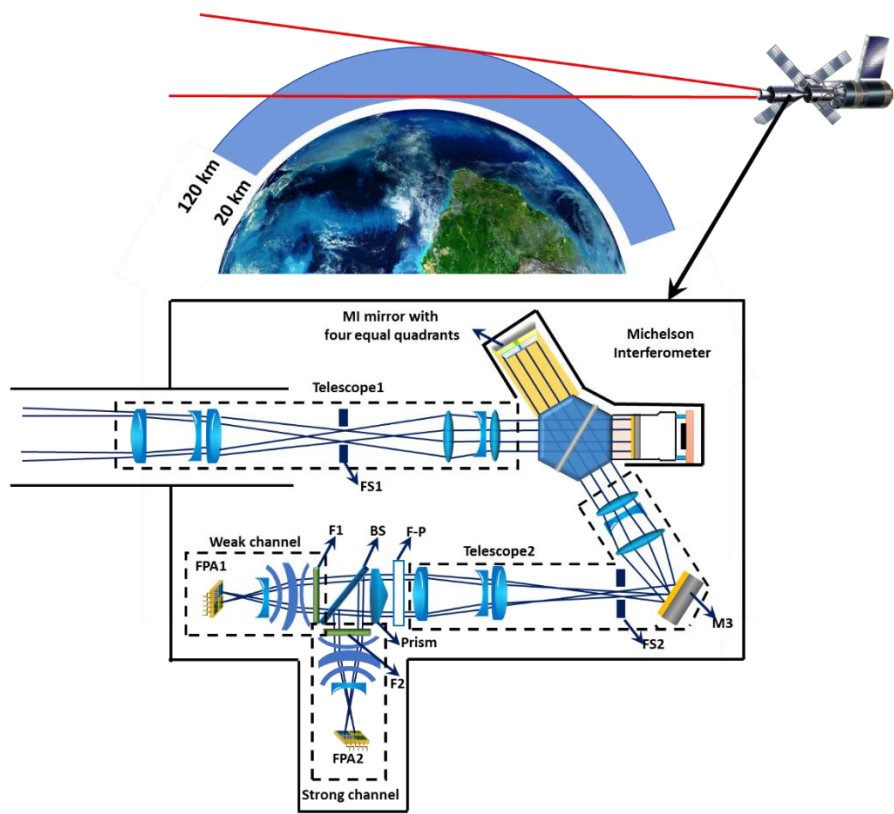

**Figure. 4: Limb-imaging geometry and optical concept schematic drawing for NWTSI.**

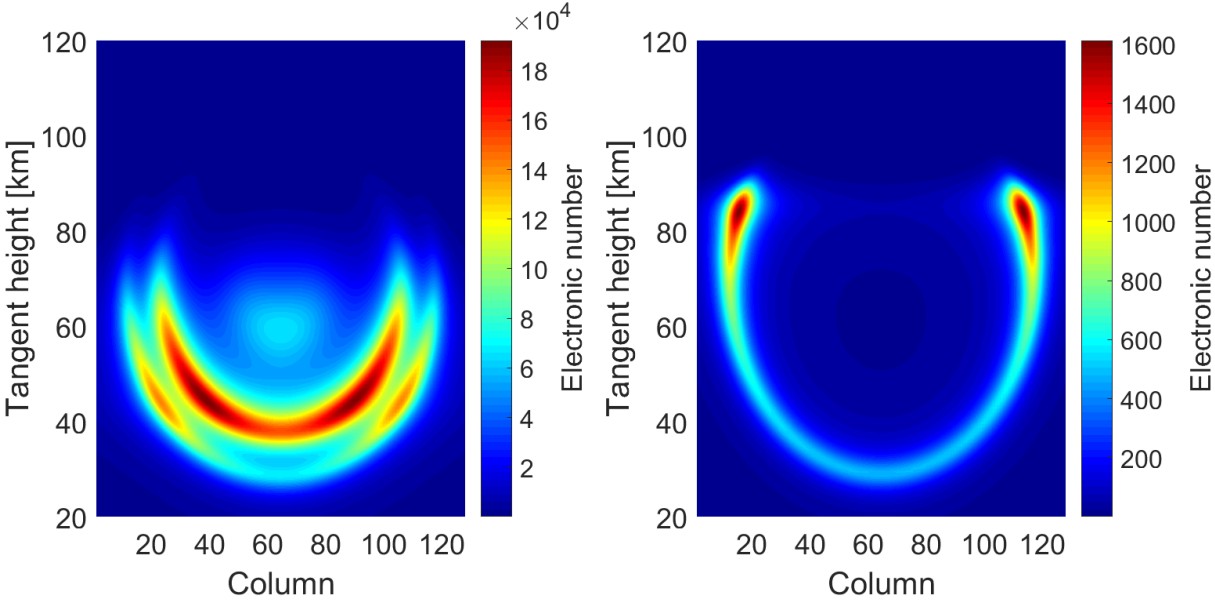

**Figure. 5: The simulated interferogram images of the weak O2(a1Δg) and OH emission lines.**





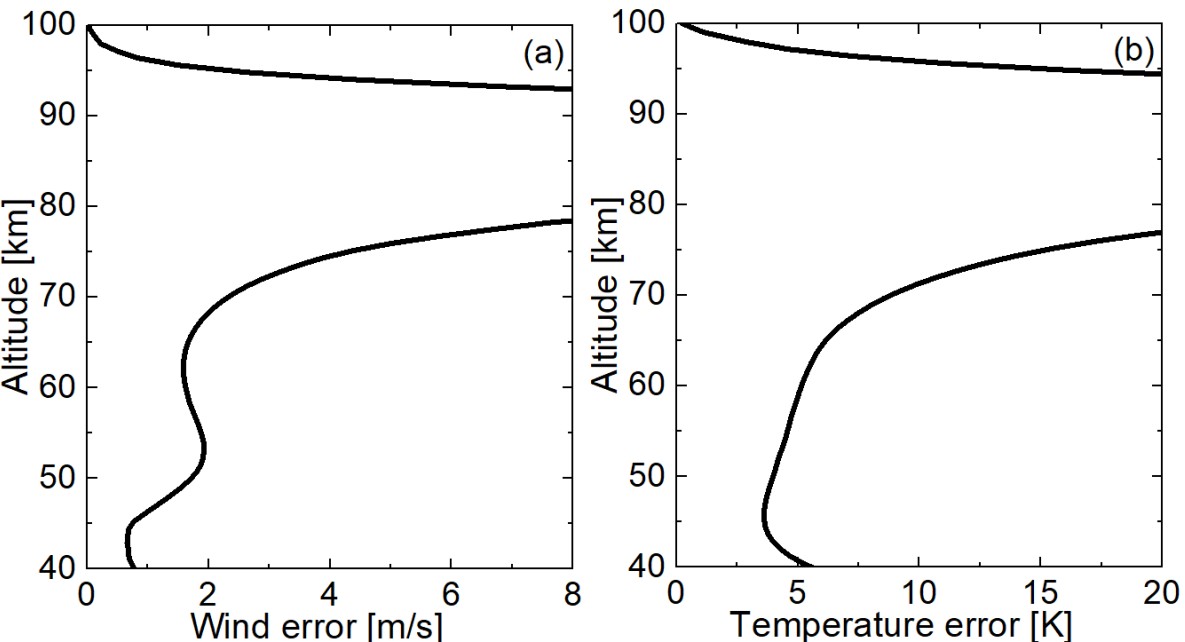

**Figure. 6: Inversion errors in wind and temperature due to Omitting the presence of OH emission.**

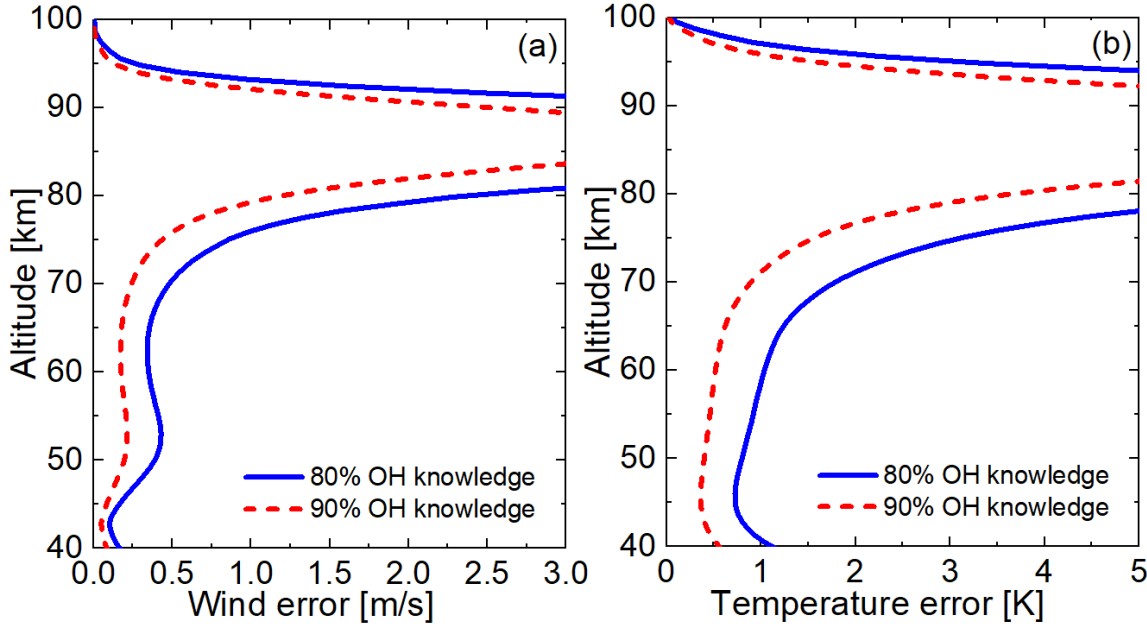

**Figure. 7: Inversion errors in wind and temperature with 80% and 90% knowledge of the OH radiance.**
