# Peer review of "Effect of OH radiance on the temperature and wind measurements derived from limb viewing observations of the 1.27 μm O₂ dayglow"

_Atmospheric Measurement Techniques, 2019_

## Referee Comment (RC1) · Anonymous Referee #1 · 12 Jul 2019

This article reports on the effect of OH atmospheric emissions on the retrievals of wind and temperature from the 1.27  micron O2 dayglow. The topic is very interesting and would deserve publication. However, I have some concern that the authors would need to consider before publishing:

MAIN CONCERNS: (1) It is written that "The OH radiance will surely affect the spectral integral intensity of the O2 mission line near 7823 cm-1 especially for altitudes between 80 to 90 km where the OH radiance is relatively strong." Given the relative lines intensities, it is difficult to believe that OH contaminate the 1.27 $\mu$m O2 dayglow. Figures 1 and 2 show that OH is not very strong. Therefore, the author should bring more

convincing arguments about the need to consider OH.

(2) The method to compute errors, with and without the knowledge of OH, is not explained in details. It could be useful to describe the method in annex.

MINOR CONCERNS:

(1) P2L33 two sets of three emission lines. What are the 2 sets exactly?

(2) Section 2.1 : Is there any auroral excitation of molecular oxygen which can lead to the 1.27 micron emission?

(3) Caption Fig 1: Should not be "Photons" instead of "Photos"?

(4) Figure 2: The caption should say what are the panels a) and b). I assume that a) is O2 while b) is OH. The caption should explain what the zoom in panel a) is. The X axis of a) and b) should be aligned, such that we see where the overlap between the lines is.

(5) It would be useful to add a plot with: Y axis: Altitude X axis: Ratio between OH and O2 VER (or radiance), in log scale. In order to see the importance (or not) of the OH lines.

(6) Figures 6/7: The errors with and without the knowledge of the OH radiance should be drawn in one plot. That will be easier to compare. It is weird not to see the full curve, i.e. around 85 km the scale is too small.
* * *

---

## Referee Comment (RC2) · Anonymous Referee #2 · 3 Dec 2019

Review of Wu O2 dayglow and OH radiance This manuscript describes the analysis the authors have made of a proposed new instrument (NWTSI) for the measurement of upper atmospheric wind and temperature, based on an earlier version (WAMI). It deals specifically with a challenge in the design arising from contamination of the primary O2 emission from the overlapping OH emission and computes the impact of this overlap on the observed winds and temperatures. The rather complex analysis is well done, and the paper is clearly written. However, I have some recommendations with respect to the nomenclature which I have listed under "Overall comments", and others which are minor but need some consideration which I have listed under "Minor comments". Overall Comments: 1. I find the units and nomenclature somewhat com-

plicated. For some reason the O2 is always "O2 dayglow" and the OH is always "OH radiance" throughout, even though they are the same thing. On line 60 we find "VER" for the first time, without explanation. The volume emission rate (VER) is the number of photons emitted from a cubic centimeter per second (see Figure 1) and is what is most widely used in the field for the airglow. Its integral (photons emitted per second from a 1 square cm column along the line of sight) is called the "integrated emission rate". Radiance is the column-integrated quantity but in milliwatts per square meter per steradian per cm-1 (wavenumber) (see Figure 2), so is similar but not the same, although it contains almost the same information, except for the spectral range. 2. The proposed new instrument contains an "ultra-narrow" filter, described in Line 138: The "ultra-narrow" filter certainly is ultra. The spectral width is not stated, but from the spectral width/free spectral range = 2.0/20 = 0.1 nm. While the fabrication of this filter/etalon is feasible for an highly skilled fabricator, it would be extremely challenging to monitor the changes in its width and central wavelength during the duration of the mission. While this comment is perhaps beyond the scope of this document, the challenge should at least be acknowledged. Minor comments: 1. On line 39, O2(a1D) should have a delta, rather than a "D". 2. On line 47, "will surely contribute" is premature to the analysis, perhaps better to use "may potentially contribute". 3. On line 89 we find "spectral irradiance", shown in Figure 2, which is close to the Integrated Emission Rate. 4. On line 95 we find "total radiance", "limb spectral radiance", "O2 dayglow" and "OH radiance". 5. Line 96: "too closed" should be "too close". 6. Line 111: Here the Michelson interferometer is described, but the Optical Path Difference is not given, which is a critical quantity in its design. 7. Line 132: Here we find FPA, but the explanation of it is missing.

8. Line 166: "ring" should be "bring". 9. Line 173: "closed" should be "close". 10. Figures 6 and 7. The plots go off scale. Aren't the off-scale values relevant?

---

## Author Response (AR1)

**Response to Reviewer 1:**

This article reports on the effect of OH atmospheric emissions on the retrievals of wind and temperature from the 1.27 μm $O_2$ dayglow. The topic is very interesting and would deserve publication. However, I have some concern that the authors would need to consider before publishing:

**Our reply:** We would like to thank the reviewer for his/her overall positive comments and the effort he/she spent on our manuscript.

**MAIN CONCERNS:**

(1) It is written that "The OH radiance will surely affect the spectral integral intensity of the $O_2$ emission line near 7823cm$^{-1}$ especially for altitudes between 80 to 90 km where the OH radiance is relatively strong." Given the relative lines intensities, it is difficult to believe that OH contaminate the 1.27 μm $O_2$ dayglow. Figures 1 and 2 show that OH is not very strong. Therefore, the author should bring more convincing arguments about the need to consider OH.

**Our reply:** Thank the Referee for the important comment and good suggestion. We have brought more convincing arguments about the need to consider OH dayglow due to its contribution to the increase of wind error and temperature error in the revised manuscript as suggested.

The total spectral irradiance including the weak group of the $O_2$ dayglow and the OH dayglow is shown in Fig. 3. The third emission line of the $O_2$ dayglow near 7823 cm$^{-1}$ is too close to the OH lines RR$_{2.5e}$ and RR$_{2.5f}$ (less than 0.05 nm) to be well optically isolated. As can be seen, the OH dayglow affects the observation of the $O_2$ mission line near 7823 cm$^{-1}$ especially for altitudes between 80 to 90 km where the OH radiance is relatively strong.

The Doppler wavenumber shift in the emission line due to wind velocity is measured as a phase shift of the interferogram, and accurate temperature measurement is determined from the ratio of the integrated absorbances of two isolated emission lines. However, the intensity variation caused by Doppler shift or temperature change is very small. The relative Doppler shift is $w/c$, where $w$ denotes the motion of the background atmosphere and $c$ is the velocity of light. If winds are to be measured to an accuracy of 3 m/s, a desirable value for the mesosphere and stratosphere, the measurement must be made to one part in $10^8$ of the velocity of light. For the central wavelength of 1270 nm, that means the measurement of the wavelength shift is about 12 fm (femtometre). Since a linewidth of the $O_2$ dayglow is of the order of 0.003 nm, the wavelength shift is $4 \times 10^{-3}$ of the linewidth. Therefore, the intensity variation of the band radiance near 7823 cm$^{-1}$ caused by the existence of the OH dayglow will surely contribute to the increase of wind error, as well as the temperature error.

(2) The method to compute errors, with and without the knowledge of OH, is not explained in details. It could be useful to describe the method in annex.

**Our reply:** Thank the Referee for the important comment and good suggestion. We have described the method to compute wind and temperature errors, with and without the knowledge of OH dayglow detailly in the revised manuscript as suggested.

The error standard deviation of inverted wind due to the presence of OH emission is found from the relation

$$\sigma_v = c \, \frac{\sqrt{J_2^2 \sigma_{J_3}^2 + J_3^2 \sigma_{J_2}^2}}{2\pi v_0 \Delta (J_2^2 + J_3^2)}$$

Where $\sigma_{J_2}^2$ and $\sigma_{J_3}^2$ represents the variance of the Fourier coefficients $J_2$ and $J_3$ due to the lack of knowledge on OH emission.

The error standard deviation of inverted temperature due to the presence of OH emission can be written as

$$\sigma_T = \frac{\Delta T}{T} = \frac{R_{AB}}{T} \frac{dT}{dR_{AB}} \sqrt{\sigma_{J_{1A}}^2 + \sigma_{J_{1B}}^2}$$

Where $\sigma_{J_{1A}}^2$ and $\sigma_{J_{1B}}^2$ represent the variance of the Fourier coefficients $J_1$ of two emission lines $A$ and $B$ of the weak group of the $O_2$ dayglow, and $R_{AB}$ is the ratio of the measured integral absorbances of this two lines, $R_{AB} = J_{1A}/J_{1B}$.

**MINOR CONCERNS:**

(1) P2L33 two sets of three emission lines. What are the 2 sets exactly?

**Our reply:** Thank the Reviewer for the very important comment. The two sets of three emission lines mentioned here refer to the weak group and the strong one shown in Fig. 1 of the sixth reference of our manuscript. We have made them clear by referring the corresponding reference in the revised manuscript.

(2) Section 2.1: Is there any auroral excitation of molecular oxygen which can lead to the 1.27 µm emission?

**Our reply:** Thank the Reviewer for the very important comment. There are auroral enhancements of the infrared atmospheric band emission ($a^1\Delta_g \rightarrow X^3\Sigma_g$) of $O_2$ near 1.27 µm. The very large aurora1 enhancements in the infrared atmospheric band of $O_2$ have been reported in the early1970s, such as the rocket measurements of Megill et al. (*J. Geophys. Res.* VOL. 75, 4775, 1970) and the observations by Noxon from an aircraft (*J. Geophys. Res.* VOL. 75, 1879, 1970). More recently, Mertens et al. (*Geophys. Res. Lett.* VOL. 35, L17106, 2008) demonstrated for the first time that $O_2^+ + NO$ charge transfer produces $NO^+(v)$. This mechanism identifies a new source of auroral infrared emission at 4.3 µm, which provides a major step forward in understanding auroral processes and a new context for understanding previously observed auroral enhancements in $O_2(a^1\Delta_g)$ band.

(3) Caption Fig 1: Should not be "Photons" instead of "Photos"?

**Our reply:** Thank the Referee for careful reading the manuscript and pointing out this problem. We have corrected this mistake in the revised manuscript as suggested.

(4) Figure 2: The caption should say what are the panels a) and b). I assume that a) is $O_2$ while b) is OH. The caption should explain what the zoom in panel a) is. The X axis of a) and b) should be aligned, such that we see where the overlap between the lines is.

**Our reply:** Thank the Referee for careful reading the manuscript and pointing out this problem. We have done this in the revised manuscript as suggested. The caption has been said what are the panels and has been explained what the zooms in the panels are. The X axis of a) and b) has been aligned.

[Figure]

Figure. 2: The spectral irradiance of the three emission lines of O2 dayglow and the two emission lines of OH dayglow at tangent heights of 40 km, 50 km, 60 km, 70 km, 80 km and 90 km. (a) three emission lines of $O_2$ dayglow. (b) two emission lines of OH dayglow. Inset to (a) or (b) shows a magnified view of a certain emission line of $O_2$ or OH dayglow, from which the linewidth and intensity varying with tangent heights can see more clearly.

(5) It would be useful to add a plot with: Y axis: Altitude X axis: Ratio between OH and $O_2$ VER (or radiance), in log scale. In order to see the importance (or not) of the OH lines.

**Our reply:** Thank the Referee for careful reading the manuscript and pointing out this problem. We have done this in the revised manuscript as suggested (please see Fig. 3(b)).

[Figure]

Figure. 3: The total spectral irradiance and band radiance as a function of tangent height. (a) the total spectral irradiance containing $O_2(a^1\Delta_g)$ and OH dayglow as a function of tangent height. (b) the band radiance profiles of the $O_2(a^1\Delta_g)$ and OH dayglow and their ratio.

(6) Figures 6/7: The errors with and without the knowledge of the OH radiance should be drawn in one plot. That will be easier to compare. It is weird not to see the full curve, i.e. around 85 km the scale is too small.

**Our reply:** Thank the Referee for careful reading the manuscript and pointing out this problem. We have done this in the revised manuscript as suggested (please see Fig. 6).

[Figure]

Figure. 6: Inversion errors in wind and temperature due to omitting the presence of OH dayglow (black curve) and with 80% and 90% knowledge of the OH dayglow (blue short dash dot and the red short dash). (a) the wind error profiles. (b) the temperature error profiles. Inset to (a) or (b) shows the wind or temperature error in the altitude range 70-100 km.

**Response to Reviewer 2:**

Review of Wu $O_2$ dayglow and OH radiance. This manuscript describes the analysis the authors have made of a proposed new instrument (NWTSI) for the measurement of upper atmospheric wind and temperature, based on an earlier version (WAMI). It deals specifically with a challenge in the design arising from contamination of the primary $O_2$ emission from the overlapping OH emission and computes the impact of this overlap on the observed winds and temperatures. The rather complex analysis is well done, and the paper is clearly written. However, I have some recommendations with respect to the nomenclature which I have listed under "Overall comments", and others which are minor but need some consideration which I have listed under "Minor comments".

**Our reply:** Thank the referee for the valuable suggestions and comments that are indispensable in improving the quality of our manuscript.

**Overall Comments:**

1. I find the units and nomenclature somewhat complicated. For some reason the $O_2$ is always "$O_2$ dayglow" and the OH is always "OH radiance" throughout, even though they are the same thing. On line 60 we find "VER" for the first time, without explanation. The volume emission rate (VER) is the number of photons emitted from a cubic centimeter per second (see Figure 1) and is what is most widely used in the field for the airglow. Its integral (photons emitted per

second from a 1 square cm column along the line of sight) is called the "integrated emission rate". Radiance is the column-integrated quantity but in milliwatts per square meter per steradian per cm-1 (wavenumber) (see Figure 2), so is similar but not the same, although it contains almost the same information, except for the spectral range.

**Our reply:** Thank the Referee for the important comment and good suggestion. We are very sorry for making a mistake of confusing expressions about "dayglow", "radiance", "VER" and "spectral irradiance". We have corrected this incorrect-used terminology and improved the clarity of those sentences including corresponding mistakes in the revised manuscript as suggested.

2. The proposed new instrument contains an "ultra-narrow" filter, described in Line 138: The "ultra-narrow" filter certainly is ultra. The spectral width is not stated, but from the spectral width/free spectral range = 2.0/20 = 0.1 nm. While the fabrication of this filter/etalon is feasible for an highly skilled fabricator, it would be extremely challenging to monitor the changes in its width and central wavelength during the duration of the mission. While this comment is perhaps beyond the scope of this document, the challenge should at least be acknowledged.

**Our reply:** Thank the Referee for the important comment and good suggestion. We have made this clear in the revised manuscript as suggested. The spectral width of the ultra-narrow filter has been added. And the challenge to monitor the changes in the width and central wavelength of the ultra-narrow filter during the duration of the mission has also been point out.

**Minor comments:**
1. On line 39, O2(a1D) should have a delta, rather than a "D".
**Our reply:** Thank the Referee for careful reading the manuscript and pointing out this problem. We have corrected this mistake in the revised manuscript as suggested.

2. On line 47, "will surely contribute" is premature to the analysis, perhaps better to use "may potentially contribute".
**Our reply:** Thank the Referee for careful reading the manuscript and pointing out this problem. We have corrected this mistake in the revised manuscript as suggested.

3. On line 89 we find "spectral irradiance", shown in Figure 2, which is close to the Integrated Emission Rate.
**Our reply:** Thank the Referee for careful reading the manuscript and pointing out this problem. In truth, $L(v)$ is "spectral irradiance". $L(v) = \int_{-\infty}^{\infty} \eta(s)D(v,s)\exp[-\int_{-s}^{\infty} n(s')\sigma(s')\mathrm{d}\,s']\mathrm{d}\,s$.

Here, the line-shape function of the emission line has been taken into account in this formula. $D(v)$ is the Doppler line-shape of a spectral line. The line-shape function $D(v)$ is normalized such that $\int D(v)\mathrm{d}v \equiv 1$. Therefore, the unit of $L(v)$ is W/m$^2$/sr/cm$^{-1}$.

4. On line 95 we find "total radiance", "limb spectral radiance", "O$_2$ dayglow" and "OH radiance".
**Our reply:** Thank the Referee for careful reading the manuscript and pointing out this

problem. We have corrected this incorrect-used terminology and improved the clarity of this sentence in the revised manuscript as suggested.

5. Line 96: "too closed" should be "too close".

**Our reply:** Thank the Referee for careful reading the manuscript and pointing out this problem. We have corrected this mistake in the revised manuscript as suggested.

6. Line 111: Here the Michelson interferometer is described, but the Optical Path Difference is not given, which is a critical quantity in its design.

**Our reply:** Thank the Referee for careful reading the manuscript and pointing out this problem. We have given the value of the Optical Path Difference in the revised manuscript as suggested.

7. Line 132: Here we find FPA, but the explanation of it is missing.

**Our reply:** Thank the Referee for careful reading the manuscript and pointing out this problem. FPA is the short for focal plane array. We have added the explanation of it in the revised manuscript.

8. Line 166: "ring" should be "bring".

**Our reply:** Thank the Referee for careful reading the manuscript and pointing out this problem. We have corrected this mistake in the revised manuscript as suggested.

9. Line 173: "closed" should be "close".

**Our reply:** Thank the Referee for careful reading the manuscript and pointing out this problem. We have corrected this mistake in the revised manuscript as suggested.

10. Figures 6 and 7. The plots go off scale. Aren't the off-scale values relevant?

**Our reply:** Thank the Referee for careful reading the manuscript and pointing out this problem. We have provided new figures in the revised manuscript as suggested (please see Fig. 6).

[Figure]

Figure. 6: Inversion errors in wind and temperature due to omitting the presence of OH dayglow (black curve) and with 80% and 90% knowledge of the OH dayglow (blue short dash dot and the red short dash). (a) the wind error profiles. (b) the temperature error profiles. Inset to (a) or (b) shows the wind or temperature error in the altitude range 70-100 km.

[revised manuscript text omitted]

$$L(v) = \int_{-\infty}^{\infty} \eta(s)D(v,s)\exp[-\int_{-s}^{\infty}n(s')\sigma(s')\mathrm{d}\,s']\mathrm{d}\,s \tag{3}$$

where $D(v)$ is the Doppler line shape of the spectral line, $\eta(s)$ is the volume emission rate $n(s)$ is the number density, $\sigma(s)$ is the absorption cross-section and $s$ is the distance along the line-of-sight.

Figure 2 shows the limb spectral radiance of three emission lines of $O_2$ and the two emission lines of OH at different altitudes (40-90 km with 10 km interval). Inset shows a magnified view of a certain emission line, from which the linewidth and intensity varying with tangent heights can see more clearly.

The total spectral irradiance including the weak group of the O2 dayglow and the OH dayglow is shown in Fig. 3(a). The total radiance including limb spectral radiance of the weak group of O₂ dayglow and the OH radiance is shown in Fig. 3. As can be seen, tThe third emission line of the $O_2$ dayglow near 7823 cm⁻¹ is too closed close to the OH lines $RR_{2.5e}$ and $RR_{2.5f}$ (less than 0.05 nm) to be well optically isolated. Figure 3(b) shows the band radiance profiles of the $O_2(a^1\Delta_g)$ near 7823 cm⁻¹ and the OH dayglow, and their ratio. As can be seen, Tthe OH dayglowradiance will surely affects the observation of spectral integral intensity of the $O_2$ emission line near 7823 cm⁻¹ especially for altitudes between 80 to 90 km where the OH radiance is relatively strong.

The Doppler shift of the emission line due to the movement of the atmosphere is measured as a phase shift of the Michelson interferometer, and accurate temperature measurement is obtained from the integrated absorbance ratio of two isolated emission lines [9]. However, the intensity variation caused by Doppler shift or temperature change is very small. The relative Doppler shift is $w/c$ , where $w$ denotes the motion of the background atmosphere and $c$ is the velocity of light. If winds are to be measured to an accuracy of 3 m/s, a desirable value for the mesosphere and stratosphere, the measurement must be made

110 to one part in $10^8$ of the velocity of light. For the central wavelength of 1270 nm, that means the measurement of the wavelength shift is about 12 fm (femtometre). Since a linewidth of the $O_2$ dayglow is of the order of 0.003 nm, the wavelength shift is $4 \times 10^{-3}$ of the linewidth. Therefore, the intensity variation of the band radiance near 7823 cm$^{-1}$ caused by the existence of the OH dayglow will surely contribute to the increase of wind error, as well as the temperature error.

**3 Forward simulation**

**3.1 The instrument model**

[revised manuscript text omitted]

$$\sigma_v = c \, \frac{\sqrt{J_2^2 \sigma_{J_3}^2 + J_3^2 \sigma_{J_2}^2}}{2\pi v_0 \Delta (J_2^2 + J_3^2)} \tag{9}$$

Where $\sigma_{J_2}^2$ and $\sigma_{J_3}^2$ represents the variance of the Fourier coefficients $J_2$ and $J_3$ due to the lack of knowledge on OH emission.

The contribution of OH dayglow to the wind error is shown in Fig. 6(a). The black curve represents omitting the presence of the OH emission. As can be seen, the wind error is about 1-2 m/s for altitudes below 70 km, climbing to greater than 8 m/s as the altitude increases from 80 to 90 km. The sensitivity of wind retrieval is weaker in the lower altitudes due to lower OH volume emission rate. The degree of knowledge of OH dayglow is important for NWTSI retrieval. The blue short dash dot and the red short dash in Fig. 6(a) represent the errors deviation of inverted wind with OH dayglow knowledge of 80 % and 90 %, respective. As can be seen, when OH dayglow is known to 80%, the wind error is 0.3–0.5 m/s for the altitude range of 40–70 km; for OH dayglow knowledge of 90%, the wind error decreases to 0.1–0.3 m/s m.

The error standard deviation of inverted temperature due to the presence of OH emission can be written as

$$\sigma_T = \frac{\Delta T}{T} = \frac{R_{AB}}{T} \frac{dT}{dR_{AB}} \sqrt{\sigma_{J_{1A}}^2 + \sigma_{J_{1B}}^2} \tag{10}$$

Where $\sigma_{J_{1A}}^2$ and $\sigma_{J_{1B}}^2$ represent the variance of the Fourier coefficients $J_1$ of two emission lines $A$ and $B$ of the weak group of the $O_2$ dayglow, and $R_{AB}$ is the ratio of the measured integral absorbances of this two lines, $R_{AB} = J_{1A}/J_{1B}$. ~~The effect of OH airglow on wind and temperature inversion can be reduced by excluding the tainted data pixels or improving OH radiance knowledge. The first option is infeasible for NWTSI because the weak emission line tainted by the OH airglow has the largest spatial coverage. Another option is to retrieve OH radiance. This option might potentially improve OH radiance knowledge to a very limited degree (depending on the uncertainty of the initial guess), and resulting OH radiance error will also be introduced by the retrieval to compensate for other error sources that are not well accounted for.~~
~~The degree of knowledge of OH radiance is important for NWTSI retrieval. Fig. 7 shows the errors deviation of inverted wind and temperature with OH radiance knowledge of 80 % and 90 %. As can be seen, when OH radiance is known to 80%, the wind error is 0.3-0.5 m/s and temperature error is 0.7-1.0 K for the altitude range of 40-70 km; for OH radiance knowledge of 90%, the wind error decreases to 0.1-0.3 m/s m and temperature error decreases to 0.3-0.5 K.~~

A similar trend with the wind error is found for the temperature obtained from $J_{1A}$ and $J_{1B}$: the temperature error is about 4-6 K for altitudes below 70 km and greater than 20 K for altitudes from 80 to 90 km (see the black curve in Fig. 6(b)). When OH dayglow is known to 80%, the temperature error is 0.7-1.0 K for the altitude range of 40–70 km (see the blue short dash dot in Fig. 6(b)); for OH dayglow knowledge of 90%, the temperature error decreases to 0.3-0.5 K (see the red short dash in Fig. 6(b)).

By happy coincidence, the wind and temperature inversion for altitude above 70 km is irrelevant to the "weak" group. Due to their relative weak self-absorption effect, the weak emission lines are used only for wind and temperature measurement at low altitude. The three strong emission lines suffering little from self-absorption at relatively higher tangent heights takes the place of the weak group for data inversion.

The performance analyses presented assume that the OH radiance is known to a degree that results in meeting the threshold wind and temperature accuracies. The largest errors are obtained when not accounting for OH radiance in the retrievals. In order to  bring down the wind and temperature error to an acceptable level, OH radiance knowledge with an accuracy of 80-90 % is required.

**5 Conclusions**

We have simulated and discussed the effect of OH radiance on the wind and temperature measurements derived from the 1.27 µm $O_2$ dayglow limb-viewing observations. We first calculated the $O_2$  and OH dayglow  spectrum over the

spectral region and altitude range of interest using the line-by-line radiative transfer model and the photochemical model incorporating the most recent spectroscopic parameters, rate constants and solar fluxes. Its shows that the OH lines $RR_{2.5e}$ and $RR_{2.5f}$ is too close to the third weak emission line of the $O_2$ dayglow near 7823 cm$^{-1}$ (less than 0.05 nm), which will surely affect the spectral integral intensity of the $O_2$ mission line. A brief description of the instrument model of NWTSI including the schematic drawing of Limb-imaging geometry and the instrument concept is presented. The method of four-point sampling of the interferogram and sample results of measurement simulations are provided for both $O_2$  and OH radiance. It was apparent from the simulations that the interference fringe of OH radiance is much stronger for pixels nearer the periphery of FPA, where the signal of the third weak emission line of the $O_2$ dayglow near 7823 cm$^{-1}$ is imaged. Inversion errors of wind and temperature measurements due to the effect of OH radiance are presented and discussed in detail. The presence of OH radiance as an interfering species decreases the NWTSI performance at all altitudes with the largest impact especially for altitudes from 80 to 90 km. The effect of OH airglow on NWTSI inversion can be reduced by improving OH radiance knowledge. Accurate $OH(v = 8)$ concentrations (uncertainty level of 20 % or better) are required to help meet the wind and temperature accuracy requirements.

*Code availability.* Available upon request.

*Author contribution.* Kuijun Wu and Weiwei He conceived the ideas, developed the forward model, performed the measurement simulation, and wrote the manuscript. Yutao Feng helped in the instrument concept and error analysis. Yuanhui Xiong calculated the atmospheric limb radiance. Faquan Li provided optical design expertise knowledge. All the authors contributed to the analysis and discussion of the results.

*Acknowledgments.* This work was supported by National Natural Science Foundation of China (NSFC) (41975039, 61705253) and National Key R&D Program of China (2017YFC0211900). The authors gratefully acknowledge informative and helpful discussions with Wang Dingyi on theoretical details of this work.

[Figure]

Figure. 7: Inversion errors in wind and temperature with 80% and 90% knowledge of the OH radiance.

315